# The Cannabinoid Receptor CB1 Stabilizes Sperm Chromatin Condensation Status During Epididymal Transit by Promoting Disulphide Bond Formation

**DOI:** 10.3390/ijms21093117

**Published:** 2020-04-28

**Authors:** Teresa Chioccarelli, Francesco Manfrevola, Veronica Porreca, Silvia Fasano, Lucia Altucci, Riccardo Pierantoni, Gilda Cobellis

**Affiliations:** 1Department of Experimental Medicine, Sez. Bottazzi, Università degli Studi della Campania “L. Vanvitelli”, Via Costantinopoli 16, 80138 Napoli, Italy; 2Department of Precision Medicine, Università degli Studi della Campania “L. Vanvitelli”, Via L. De Crecchio 7, 80138 Napoli, Italy

**Keywords:** estrogens and endocannabinoids, caput and *cauda* epididymis, type-1 cannabinoid receptor (CB1), sperm chromatin condensation and maturation, protamine thiol oxidation, chromatin remodeling and histone displacement, spermiogenesis, post-translational modification of histone H4 (histone H4 acetylation), chromodomain protein CDYL, bromodomain protein BRDT

## Abstract

The cannabinoid receptor CB1 regulates differentiation of spermatids. We recently characterized spermatozoa from *caput* epididymis of *CB1*-knock-out mice and identified a considerable number of sperm cells with chromatin abnormality such as elevated histone content and poorly condensed chromatin. In this paper, we extended our findings and studied the role of CB1 in the epididymal phase of chromatin condensation of spermatozoa by analysis of spermatozoa from *caput* and *cauda* epididymis of wild-type and *CB1*-knock-out mouse in both a homozygous or heterozygous condition. Furthermore, we studied the impact of *CB1*-gene deletion on histone displacement mechanism by taking into account the hyperacetylation of histone H4 and players of displacement such as Chromodomain Y Like protein (CDYL) and Bromodomain testis-specific protein (BRDT). Our results show that CB1, via local and/or endocrine cell-to-cell signaling, modulates chromatin remodeling mechanisms that orchestrate a nuclear condensation extent of mature spermatozoa. We show that *CB1*-gene deletion affects the epididymal phase of chromatin condensation by interfering with inter-/intra-protamine disulphide bridges formation, and deranges the efficiency of histone removal by reducing the hyper-acetylation of histone H4. This effect is independent by gene expression of *Cdyl* and *Brdt* mRNA. Our results reveal a novel and important role for CB1 in sperm chromatin condensation mechanisms.

## 1. Introduction

Spermiogenesis is the terminal differentiation phase of spermatogenesis by which round spermatids (SPTs) undergo impressive histomorphological changes that facilitate development of mature and elongated cells carrying tightly condensed chromatin and nuclei, namely mature SPTs or spermatozoa (SPZs). When spermiogenesis completes, spermiation promotes detachment of SPZs from Sertoli cells. Downstream, a contractile tubular propulsion leads SPZs from tubular lumen-to-rete testis until *caput* region of epididymis. Sperm maturation occurs during epididymal transit from the *caput*-to-*cauda* region [1].

Chromatin condensation extent of mature SPZs is orchestrated by testicular and epididymal events. These require chromatin remodeling mechanisms such as histone displacement/protamination and inter/intra-protamine disulphide bonds formation, respectively [2]. In developing germ cells, nuclear condensation is mainly related to i) haploid expression of transition proteins (TNP1 and TNP2) and protamines, ii) histone post-translational modifications (PTMs) and displacement, and iii) histone-to-protamine exchange and DNA packaging [3,4,5,6]. The combined histone H4 acetylation at lysine (K) residues K5, K8, K12, and K16 results in the main signal of global histone removal as the Bromodomain testis-specific protein (BRDT) reads and binds acetyl lysine eliciting histone displacement [7]. Additional histone PTMs are involved in this process [8]. The histone crotonylation is a new histone PTMs recently characterized in mouse germ cells [9,10]. A significant histone hyper-crotonylation has been described in elongating SPTs. This is responsive to down-regulation of Chromodomain Y Like protein (CDYL) and it has been related to histone removal. CDYL is a chromodomain protein that regulates negatively histone lysine crotonylayion (Kcr) because of its activity on crotonyl donor as crotonyl-CoA hydratase [10]. CDYL activity counteracts the acetyl-lysine reader BRDT since recent studies suggest that most bromodomains do not read cronyl-lysine [11,12]. In the *Cdyl* transgenic mouse model, the overexpression of CDYL decreases histone Kcr in elongating SPTs and interferes with histone displacement, which reveals a key role of CDYL in spermiogenesis as a modulator of histone PTMs with functional implications in histone removal mechanism [10].

In mammals, histone displacement preserves a small percentage of chromatin condensed by histones (2%–5% in mouse, 10%–15% in human) so that SPZs contain nucleoprotamines and a small fraction of nucleohistone chromatin [13,14]. Any interference with histone displacement in SPTs interferes with histone/protamine content and chromatin condensation of SPZs.

Protamines are sperm-specific nuclear proteins with high DNA affinity, as these are small and highly-basic proteins with an arginine-rich core. In eutherian mammals, including mouse and human, protamines are characterized by an arginine and cysteine residues [15]. During spermiogenesis, arginine residues mediate formation of highly stable DNA-protamine complexes that strongly condense chromatin in the toroidal structures [16]. During post-testicular maturation, cysteine residues close chromatin in tighter arrangement of protamines organizing toroids.

During epididymal transit, from *caput*-to-*cauda*, SPZs acquire their potential to move [17] and further condense chromatin through intra-/inter-protamine disulphide bonds [18] useful to stabilize and strongly condense DNA. Protamines have been recently proposed to be involved in imprinting and epigenetic regulation of sperm cells [15,19,20].

Signals and pathways involved in physiologic sperm thiol oxidation in the epididymis are not completely understood. However, inter-/intra-protamine disulphide bridges formation is a crucial step, as this strongly condense nuclei of SPZ in a functionally and completely mature state that preserves genome integrity. The highly condensed chromatin masks the DNA from stressing agents, while uncondensed chromatin is mainly associated with DNA damage [21].

All the events previously described are finely regulated by neuroendocrine axis and local factors that act through endocrine, autocrine, and paracrine pathways [2,22]. Among factors, endocannabinoids have an emerging role because of their activity at hypothalamic, testicular, and epididymal level [22].

Endocannabinoids are cell-to-cell signaling mediators able to bind the cannabinoid receptor type-1 and type-2 (CB1 and CB2) which are highly expressed in testis. Anandamide (AEA) and 2-arachidonoylglycerol (2-AG) are the main endocannabinoids characterized in testis and epididymis with a key role in spermatogenesis and sperm motility acquisition, respectively [22,23,24]. CB1 has been related to somatic and germ cell activities [24,25,26,27,28,29,30,31,32,33,34,35] and, in particular, to Leydig cell differentiation [30], steroidogenesis [25,35,36,37], spermiogenesis [2,38], quality, and epididymal maturation of SPZs [24,27,28,29,33,34,39,40].

Recent information about CB1 involvement in spermiogenesis came from male mice carrying a *CB1*-*null* deletion, either under heterozygous (CB1^+/-^) or homozygous conditions (CB1^-/-^) [33,41]. The CB1^-/-^ mice show down regulation of hypothalamus-pituitary-gonad axis with low plasma levels of testosterone and 17β-Estradiol (E_2_) [25,33,38]. These animals produce SPZs with mature and immature chromatin (condensed and uncondensed, respectively) because of heterogeneous histone content [2,33,38], likely ascribed to inefficient histone removal during spermiogenesis. 

We recently characterized SPZs from *caput* epididymis of CB1^-/-^ mice and identified a considerable number of SPZs with a chromatin abnormality such as elevated histone content, poorly condensed chromatin, highly damaged DNA, and elongated nuclear size [2,33,34,38]. We showed that all these abnormalities were correlated to each other and responsive to down-regulation of neuroendocrine axis supporting gonadotropin-E_2_ production since E_2_-treated CB1^-/-^ mice restored the number of SPZs with chromatin abnormalities to physiological values, which suggests the hypothesis that sperm chromatin quality was responsive to neuroendocrine activity of CB1 via E_2_-mediated mechanism [33,34]. Sperm chromatin of CB1^+/-^ mice appeared more similar to WT than CB1^-/-^ mice.

In this study, we extended our findings and analyzed the regulatory activity of CB1 in epididymal phase of sperm chromatin condensation. In particular, using wild-type (CB1^+/+^ or WT) and *CB1-*deleted mice, both in a homozygous and condition, we studied sperm chromatin condensation status during the epididymal transit, by comparative analysis of SPZs from *caput* and *cauda* epididymis. Furthermore, we characterized a deficit of intra-testicular E_2_ levels and signal associated to inefficient histone displacement in CB1^-/-^ mice.

## 2. Results

### 2.1. Effects of CB1 Deletion on Sperm Chromatin Condensation During the Epididymal Transit

Sperm samples from *caput* and *cauda* epididymis of WT, CB1^+/-^ and CB1^-/-^ mice were stained with Acridine Orange (AO) dye in acid conditions and comparatively analyzed by flow cytometry. The percentage of SPZs with high DNA stainability (i.e., HDS) or susceptibility of DNA to acid denaturation at strand break (i.e., DD), and thiol/disulphide status (i.e., TDS) were evaluated and used as spermatic indices of uncondensed chromatin, DNA denaturation and damage, and thiol groups oxidation status, respectively. All the HDS and TDS values, independently by genotype and epididymal region, were used for correlation analysis.

Figure 1A shows histograms of green- (FL1-H) and red- (FL3-H) stained *caput* and *cauda* SPZs from WT, CB1^+/-^, and CB1^-/-^ mice in the gated areas (M1 and M2).

Figure 1 shows the relative HDS and DD values (panel B and C) while Figure 2 show the relative TDS values (panel A) and correlation analysis between HDS and TDS values (panel B).

Sperm samples from *caput* and *cauda* epididymis of WT and CB1^+/-^ mice showed scanty and comparable HDS values with no significant differences *caput* vs. *cauda* (Figure 1B). Sperm samples from *caput* and *cauda* epididymis of CB1^-/-^ mice showed HDS values significantly higher compared to WT and CB1^+/-^ (*p* < 0.01), both in *caput* and *cauda* epididymis with a significant increase, *caput-*to-*cauda* (*p* < 0.05).

Both in *caput* and *cauda* epididymis, sperm cells showed DD values significantly higher in CB1^-/-^ mice than in WT and CB1^+/-^ (*p* < 0.01) (Figure 1C). However, in WT and CB1^+/-^, SPZs showed DD values higher in *caput* than in *cauda* epididymis (*p* < 0.01), while, in CB1^-/-^ mice, SPZs showed comparable DD values in *caput* and *cauda* epididymis.

In WT and CB1^+/-^, SPZs showed TDS values significantly higher in *caput* than in *cauda* epididymis (*p* < 0.01), while in CB1^-/-^ mice, SPZs showed TDS values with no significant difference *caput* vs. *cauda* (Figure 2A). In *caput* epididymis, SPZs showed TDS values significantly higher in CB1^+/-^ mice when compared to WT (*p* < 0.01) with a further significant increase in CB1^-/-^ mice (*p* < 0.01). In *cauda* epididymis, SPZs showed scanty and comparable TDS values in WT and CB1^+/-^ mice, and higher TDS values in CB1^-/-^ mice compared to WT and CB1^+/-^ mice (*p* < 0.01).

Correlation analysis shows that HDS and TDS values are positively (*r* = 0.82) and significantly (*p* < 0.01) correlated with each other (Figure 2B).

### 2.2. Impact of CB1-Gene Deletion on Chromatin Condensation of Efficiently Protamined CB1-Null SPZs During the Epididymal Transit

E_2_ treatment was used as an experimental approach to recover sperm protamination efficiency in *CB1*-*null* mice. This experimental approach was useful to study the impact of *CB1*-gene deletion on sperm protamine thiol group oxidation during the epididymal transit by avoiding interference by poor protamination.

CB1^-/-^ mice were treated with vehicle or E_2_ ± ICI and analyzed in comparison to WT mice, which are used as a control reference in this case. Efficiency of histone displacement was used as an experimental control of E_2_-treatment efficacy and evaluated by analysis of testicular content of histone H3. Spermatozoa collected from *caput* and *cauda* epididymis were used to analyze the chromatin condensation status via HDS and TDS indices.

Histone H3 (Figure 3A) was present in testis, independently by genotype or treatment (upper panel). Quantitative densitometry analysis of signals (lower panel) showed that histone H3 levels were significantly lower in E_2_-treated CB1^-/-^ mice in comparison to other experimental groups (*p* < 0.05). No significant difference between Control (CTRL) vs. E_2_ + ICI was observed. E_2_-treatment restored histone H3 levels to WT values. This rescue activity demonstrated the efficiency of E_2_ treatment and confirmed that E_2_-treated CB1^-/-^ mice release in *caput* epididymis efficiently protamined SPZs because of histone displacement recovery [38].

Figure 3B show HDS values. In WT mice, SPZs showed scanty and comparable HDS values in both *caput* and *cauda* epididymis. In all the experimental groups, SPZs showed HDS values significantly lower in *caput* than in *cauda* epididymis (*p* < 0.01) with comparable values in *cauda*. Comparatively to CTRL or E_2_ + ICI groups, E_2_-treatment significantly reduced HDS values in *caput* epididymis (*p* < 0.01), which confirms efficiency of E_2_ treatment [33]. In WT mice and E_2_ treated animals, *caput* SPZs showed similar HDS values.

Figure 3C show TDS values. In WT mice, SPZs showed TDS values higher in *caput* than in *cauda* epididymis (*p* < 0.01). In all the experimental groups, SPZs showed comparable TDS values in both *caput* and *cauda* epididymis. Comparatively to CTRL or E_2_ + ICI groups, E_2_ treatment significantly reduced sperm TDS values both in *caput* and *cauda* epididymis (*p* < 0.01). In *caput* epididymis, SPZs showed similar TDS values in WT and E_2_-treated animals.

### 2.3. Characterization of Intra-testicular E_2_ Deficit and Histone PTMs Associated to Inefficient Histone Displacement in CB1^-/-^ Mice

Using WT and CB1^-/-^ mice, we verified intra-testicular deficit of E_2_ levels associated to inefficient histone removal and studied the molecular mechanism of histone displacement by taking into account histone H4 acetylation.

In particular, using WT and CB1^-/-^ mice, we analyzed the gene expression of *Cdyl* and *Brdt*, as a significant player in histone removal, and analyzed the histone H4tetraAc levels, as a main signal of histone displacement [8]. Histone H3 was used for monitoring efficiency of histone displacement and normalized histone H4tetraAc levels. We excluded analysis of CB1^+/-^ mice as a sperm phenotype of these animals, at molecular and biochemical levels, including histone retention, is very similar to that of WT mice [33,38].

Results demonstrate that E_2_ were significantly (*p* < 0.01) lower in CB1^-/-^ (0.73 ± 0.043 pg/mg) than in WT (1.24 ± 0.049 pg/mg) testis (Figure 4), which revealed ~40% of E_2_ reduction.

Gene expression analysis of *Cdyl* and *Brdt* showed no significant differences, WT vs. CB1^-/-^ (Figure 5A), while significant variation was observed when we analyzed histone H3 and H4tetraAc levels. Quantitative densitometry analysis of signals showed a significantly higher histone H3 content in CB1^-/-^ than in WT mice both in testis (*p* < 0.01) and in *caput* SPZ (*p* < 0.05). Relatively to total histone content, histone H4tetraAc was higher in WT compared to CB1^-/-^ testis (Figure 5B,C).

## 3. Material and Methods

### 3.1. Experimental Animals

CB1-wild-type (CB1^+/+^ or WT) male mice or males carrying a CB1 null mutation [41] either under heterozygous (CB1^+/-^) or homozygous (CB1^-/-^) conditions were used in this study. Heterozygous mice were bred on a CD1 background (Charles River Laboratory, Lecco, Italy) before generating male mice (WT, CB1^+/-^, CB1^-/-^).

The number of the enrolled adult animals was determined by the parameters that we have to adopt for the G*Power analysis required to get the permission for in vivo experiments in Italy, which is suggested by the Legal Entity giving the permission.

All animals were maintained on a standard pellet diet with free access to water. Adult males (4–8 months) were killed by CO_2_ asphyxia and testes and/or epididymis were processed depending on the experimental procedure, as previously described [38]. In detail, testes were rapidly removed and properly stored at −80 °C, while epididymis were dissected and used to collect SPZs from *caput* (*caput* SPZ) and *cauda* (*cauda* SPZ) regions, as described below.

Experiments were approved by the Italian Ministry of Education and the Italian Ministry of Health with authorization n°941/2016-PR issued on 10.10.2016. Procedure involving animal care were carried out in accordance with the National Research Council’s publication *Guide for Care and Use of Laboratory Animals* (National of Institutes of Health Guide).

### 3.2. In Vivo Experiment with 17-β Estadiol (E_2_)

CB1^-/-^ male mice (*n* = 11) of 24 days postpartum (*dpp*) were injected with physiological solution containing vehicle (1% ethanol, *n* = 3 animals), E_2_ (1.5 μg/100 g dose for each injection, *n* = 4 animals), E_2_ (1.5 μg/100 g dose for each injection) in combination with the estrogen receptor (ER) antagonist ICI182780 (ICI, 15 μg/100 g dose for each injection, *n* = 4 animals for E_2_+ICI group). We decided to avoid the anti-estrogen alone from analysis due to its effect on testis, efferent ductile, and epididymis [42]. All the substances were dissolved in ethanol, diluted in 100 μL physiological solution (each injection/dose contained 1% ethanol), and injected intraperitoneally on alternate days for seven weeks. Doses and time of treatment have previously been reported [2,33].

In adult animals, overlapping waves of spermatogenesis continuously produce SPZs that accumulate in the *cauda* region of the epididymis. Therefore, in order to recover and analyze the largest and most homogeneous number of efficiently protaminated SPZs, the pharmacological treatment was performed on prepubertal animals during the first wave of spermatogenesis. Specifically, treatment was performed on 24 *dpp* mice according to the presence of round SPT and was halted seven weeks later because the first wave of spermatogenesis and sperm transit along epididymis, from *caput*-to-*cauda*, require approximately 60 days. At the end of this period, WT animals (70-days-old) and all treated animals were euthanized with CO_2_ and testes or epididymis (*n* = 3 or 4 for experimental group) were processed depending on the experimental procedure. In detail, testes were rapidly removed, fixed, or frozen on dry ice and properly stored for routine histological control and Western blot analysis. Epididymis were dissected and used to collect *caput* and *cauda* SPZs, as described below.

### 3.3. Sperm Collection From Caput and Cauda Epididymis

*Caput* and *cauda* epididymis (from *n* = 4 mice for each genotype) were separately immersed in PBS (pH 7.6) and cut to let SPZs flow out from the ducts, as previously reported [38]. Samples of *caput* and *cauda* SPZs were then filtered throughout cheesecloth, dissolved in 1 ml of ice-cold PBS (pH 7.4) buffer (1 × 10^6^/100μL), and centrifuged at 600xg for 5 min. Aliquots of *caput* SPZ from WT and CB1^-/-^ mice were frozen on dry ice and stored for Western blot analysis (*n* = 10 WT aliquots and 10 CB1^-/-^ aliquots from each of animals). Aliquots of *caput* and *cauda* SPZs from WT, CB1^+/-^ and CB1^-/-^ mice were properly fixed [38] and processed for AO staining (*n* = 4 WT aliquots and 4 CB1^-/-^ aliquots from each of animals).

### 3.4. Acridine Orange Staining Analysis

The fluorochrome AO intercalates into double strand DNA (native DNA) as a monomer and fluoresces green. Conversely, when it binds to single strand DNA (denatured or single strand DNA) as an aggregate, a red fluorescence is observed. DNA is vulnerable to denaturation under acid conditions [38,43]. This metachromatic shift from green (FL1-H) to red (FL3-H) has been used to measure chromatin quality indices of *caput* and *cauda* SPZs under acid conditions [44,45]. In particular, using cytofluorimetry analyses, we evaluated the percentage of SPZs with high DNA stainability (i.e., HDS) or susceptibility of DNA to acid denaturation at strand break (i.e., DD), and thiol/disulphide status (i.e., TDS). Values were considered as spermatic indices of uncondensed chromatin (i.e., HDS, calculated as intensely green [FL1-H >10^3^] fluorescing DNA/total fluorescing DNA [FL1-H >10^1^ + FL3-H > 10^1^]), DNA denaturation and damage (i.e., DD, calculated as red fluorescing [FL3-H > 10^1^]/total fluorescing DNA, [FL1-H >10^1^ + FL3-H > 10^1^]), and thiol groups oxidation status (i.e., TDS, calculated as red fluorescing [FL3-H > 10^1^]/green fluorescing [FL1-H >10^1^] DNA), respectively [33,38,44,45].

Sperm samples were treated for 30 s with 400 μL of a solution of 0.1% Triton X-100, 0.15 M NaCl and 0.08 N HCl, pH 1.2. After 30 s, 1.2 mL of staining buffer [6μg/mL AO, 37 mM citric acid, 126 mM Na_2_HPO_4_, 1mM disodium EDTA, 0.15 M NaCl, pH 6.0] was admixed to the test tube and analyzed by flow cytometry. After excitation by a 488 nm wavelength light source, AO bound to a double-stranded DNA fluorescent green (515–530 nm) and AO bound to a single-stranded DNA fluorescent red (630 nm or greater). A minimum of 10,000 cells were analyzed by fluorescent activated cell sorting (FACS Calibur, BD BioScience, Milan, Italy).

### 3.5. RT-qPCR

Testes from WT and CB1^-/-^ mice (*n* = 6 WT testes and 7 CB1^-/-^ testes) were homogenized in TRIZOL Reagent (Invitrogen Life Technologies, Milan, Italy) in agreement with manufacturer’s instructions for total RNA extraction. RNA aliquots (10 µg) were treated with 1 µl Deoxyribonuclease (DNAse, 10 U/µl) at 37 °C for 10 min and processed in agreement with manufacturer’s instructions (GE Healthcare, Milan, Italy). Purity and integrity of RNA samples (*n* = 3 WT and 3 CB1^-/-^) were determined by spectrophotometry and electrophoresis. Then RNA was used for cDNA synthesis and real time-qPCR, as already reported [46]. Gene expression analysis, corrected for PCR efficiency and normalized toward a reference gene [ribosomal protein S18, Rps18 (sense: 5′-gagactctggcatgctaactag-3′; antisense: 5′-ggacatctaagggcatcacag-3′)], was performed by CFX Manager software (Bio-Rad). Normalized fold expression of mRNAs was calculated by applying the 2^-ΔΔCt^ method.

Primer sequences for *Cdyl* mRNA (sense: 5′-gaaagcactaaaatggcagac-3′, antisense: 5′-gcccaaaccacatcacaaag -3′) and *Brdt* mRNA (sense: 5′-gattccttctgggcttcctg-3′, antisense: 5′-gctatggaagtggtaggagtt-3′) were designed through Primer3 (http://primer3.ut.ee/).

Results were expressed as a mean value of normalized fold expression (nfe) ± SEM.

### 3.6. Western Blot Analysis

Testes (*n* = 6 WT testes and 7 testes CB1^-/-^) or *caput* SPZs (*n* = 6 WT aliquots and 7 CB1^-/-^ aliquots), were homogenized in RIPA buffer (PBS, pH 7.4, 10 mM dithiothreitol, 0.02% sodium azide, 0.1% SDS, 1% Nonidet P-40, 0.5% sodium deoxycholate) in the presence of protease inhibitors (10 µg/mL of leupeptin, aprotinin, pepstatin A, chymostatin, and 5 µg/mL of TPCK), as already reported [33] and analyzed by Western blot. Briefly, proteins (10 μg for testis and 5 μg for *caput* SPZ) were separated by SDS-PAGE (15% acrylamide) and transferred to polyvinylidene difluoride membrane (GE Healthcare) at 280 mA for 2.5 h at 4° C. Membrane was cut at 25kDa level. The upper and lower filters were treated for 3 h with blocking solution [5% nonfat milk, 0.25% Tween-20 in Tris-buffered saline (TBS, pH7.6)] and then separately incubated overnight at 4° C in TBS-milk buffer (TBS pH 7.6, 3% nonfat milk) with the primary antibody (Total Histone H3, diluted 1:1000, code 05-928 from Merck Millipore, Burlington, USA, ACTIN, diluited 1:5000, code E-AB-20034, Elabscience Biotechnology, Wuhan, China and ERK2, diluted 1:1000, code sc-1647 from Santa Cruz Biotechnology, Inc., Heidelberg, Germany, H4tetraAc diluted 1:1000, code 05-1355, from Merck Millipore, Burlington, USA). After washing in 0.25% Tween20-TBS, filters were incubated with 1:1000 horseradish peroxidase-conjugated rabbit or mouse IgG (Dako Corp., Milan, Italy) in TBS-milk buffer and then washed again. The immune complexes were detected using the enhanced chemiluminescence-Western blotting detection system (Amersham ECL Western Blotting Detection Reagent, cod: RPN2106, GE Healthcare). Signals were quantified by densitometry analysis, appropriately normalized relatively to levels of ERK1/2, ACTIN, or total H3 (post-translationally modified and unmodified H3), and graphed as a fold change of Optical Density (OD) (mean ± S.E.M.). Since, in sperm protein extract, the ERK2 signal quickly reached saturation. To limit the error, we used ACTIN as a loading control.

Specificity of the immunoreactions was already demonstrated [24,30,47,48] and, in this scenario, routinely checked by omitting the primary antibody.

### 3.7. 17-β Estadiol by Enzyme Immunoassay (EIA) 

Testes from WT and CB1^-/-^ mice (*n* = 6 WT mice and 6 CB1^-/-^ mice) were homogenized in 70% methanol and extracted with 2 × 7 mL diethyl ether. After drying, each extract (*n* = 6 WT and 6 CB1^-/-^) was dissolved in phosphate buffered saline (PBS 0.1 M, pH 7) containing 0.2% gelatine and used for E_2_ determination [49].

Intra-testicular E_2_ levels were quantified using a commercially available Enzyme Immunoassay (EIA) kit, according to the manufacturer’s instructions (Cayman, Florence, Italy). The kit was a competitive assay recommended for quantification of E_2_ from tissue with a detection limit of approximately 20 pg/mL. Different concentrations of E_2_ (Sigma-Aldrich) were used to verify sensibility of the assay. The intra-assay and inter-assay coefficient of variation have been determined at multiple points on a standard curve resulting as 7.9% and 5.3%, respectively. Each extract (*n* = 6 WT extracts and *n* = 6 CB1^-/-^ extracts from different animals) was analyzed in triplicate and values were expressed in pg for mg of tissue (pg/mg) and graphed as mean ± S.E.M.

### 3.8. Correlation Analysis

All the HDS and TDS values relative to *caput* and *cauda* SPZ of WT, CB1^+/-^, and CB1^-/-^ mice have been correlated with each other. Specifically, we included in our analysis both *caput* and *cauda* SPZ without considering the genotype and epididymal region. Data have been compared using the Excel built-in distribution functions available in Microsoft Office. The value of r was considered to establish the test significance. The range −1≤ r ≤ 1 established a negative or positive correlation between HDS and TDS.

### 3.9. Data Presentation and Statistical Analysis

ANOVA followed by Duncan’s test (for a multi-group comparison) or Student’s t-test (for two independent group comparisons) were conducted to identify groups with a different mean. Data were expressed as the mean ± S.E.M. from at least three independent animals for each genotype or experimental group. For RTqPCR and Western blot analyses, triplicates from 4–7 animals/genotypes or experimental groups were considered.

## 4. Discussion

The chromatin condensation extent of mature SPZs is mainly orchestrated by testicular and epididymal events that involve chromatin remodeling mechanisms such as histone displacement/protamination and inter/intra-protamine disulphide bonds formation, respectively.

Recently, we studied the impact of *CB1* deletion on histone displacement by comparing histone retention and chromatin condensation of SPZs isolated from *caput* epididymis of WT, CB1^+/-^, and CB1^-/-^ mice [2,33,34,38]. In this case, we extend our previous findings and studied the impact of *CB1* deletion on the epididymal event that underlies chromatin packaging of SPZs transiting along epididymis, by comparing chromatin condensation indices of SPZs isolated from *caput* and *cauda* epididymis of WT, CB1^+/-^, and CB1^-/-^ mice. Specifically, the percentage of SPZs with high DNA stainability (i.e., HDS) and thiol/disulphide status (i.e., TDS) were evaluated in sperm samples collected from *caput* and *cauda* of epididymis. Values were considered as spermatic indices of uncondensed chromatin and thiol groups’ oxidation status (i.e., disulphide brides formation), respectively. DNA damage is strongly related to uncondensed chromatin. Therefore, we also analyzed the susceptibility of DNA to acid denaturation at a strand break, which is considered as spermatic indices of DNA denaturation and damage (i.e., DD). However, the need of the present study was also to fulfill some gaps of knowledge about the neuroendocrine role of CB1 in the histone displacement mechanism. Therefore, using WT and CB1^-/-^ mice, we characterized the intra-testicular deficit of E_2_ levels associated with inefficient histone removal and also studied the molecular mechanism of histone displacement.

Our results suggest that CB1 activity preserved the chromatin condensation status of SPZ transiting along the epididymis [2,33,34,38]. Indeed, the comparative analysis of sperm samples collected from *caput* and *cauda* epididymis of WT and CB1^+/-^ mice revealed scanty HDS values, with no significant difference *caput* vs. *cauda*. Such a phenotype was reasonably ascribable to the efficiency of the DNA protamination process occurring during spermiogenesis since protamine-based structures are reported to highly condense chromatin of SPZs. Either WT or CB1^+/-^ mice mostly produce efficiently protaminated SPZs with a physiological content of histone proteins [38]. However, the comparative analysis of sperm samples collected from *caput* and *cauda* epididymis of CB1^-/-^ mice showed HDS values were significantly high compared to WT and CB1^+/-^ mice both in *caput* and *cauda*, with a significant increase from *caput*-to-*cauda*, which suggests that CB1 gene deletion, besides increasing the production of poorly condensed SPZs in *caput* epididymis [38], also interfered with sperm chromatin maturation during epididymal transit. The observed susceptibility of chromatin to swelling detected in SPZs transiting from *caput*-to-*cauda* suggested that *CB1* deletion, more than poorly protaminated DNA, negatively affected the sperm chromatin condensation status during the epididymal transit. Data on the susceptibility of DNA to acid denaturation and thiol-disulphide status confirmed our hypothesis and revealed a new effect of CB1 deletion on epididymal maturation of SPZs.

In *caput* and *cauda* epididymis, sperm cells showed DD values are significantly higher in CB1^-/-^ mice than in WT and CB1^+/-^. However, DD values decreased from *caput* to *cauda* epididymis, both in WT and CB1^+/-^ mice, while comparable DD values were observed in *caput* and *cauda* epididymis of CB1^-/-^ mice, which indicates genotype-dependent effects on DNA denaturation and damage of SPZs transiting along the epididymis [38]. Using the alkaline Comet assay as a highly sensitive test to reveal the percentage of damaged DNA, at a single and a double strand, we previously demonstrated that DNA damage increased in mouse SPZ transiting from *caput* to *cauda* [38]. This occurred independently by genotype, and, at a smaller extent, in WT and CB1^+/-^ compared to CB1^-/-^. However, DD values used as the index of DNA denaturation and damage in this scenario specifically measure susceptibility of DNA to acid denaturation at sites of DNA strand breaks [50]. When disulphide bridges-rich protamines condenses chromatin, the DNA molecule is resistant to acid denaturation since disulphide bridges counteract DNA melting [51]. Consequentially, in *cauda* epididymis more than in *caput*, the observed DD values likely reflected the protamine thiol oxidation, which is lower in SPZs of CB1^-/-^ than WT and CB1^+/-^ mice. Accordingly, TDS decreased in SPZs transiting from *caput*-to-*cauda* epididymis both in WT and CB1^+/-^ mice, while, in CB1^-/-^ mice, it was stably elevated in SPZs transiting from *caput* to *cauda*. This suggests that CB1 gene deletion interfered with thiol oxidation. However, disulphide bridges’ formation in sperm cells is not exclusively restricted to protamines. Several events related to sperm maturation (e.g., progressive motility acquisition, fertilizing ability) require thiol oxidation of sperm proteins [52]. To confirm that the TDS values observed in *caput* and *cauda* SPZs of WT, CB1^+/-^ and CB1^-/-^ mice were ascribable to protamines, we carried out correlation analysis between TDS and HDS values, independently, by genotype and the epididymal region. The correlative analysis confirmed that thiol groups oxidation was significantly and directly associated to condensation of protamine-based chromatin indicating that the observed TDS values were related to protamines more than other proteins. DNA denaturation and chromatin de-condensation of SPZs can be originated by abnormal protaminization and/or by anomalies in the epididymal maturation. To corroborate the emerging idea that the observed susceptibility of chromatin to swelling detected in SPZs transiting along epididymis of CB1^-/-^ mice was related to an interfering effect of *CB1* deletion on disulphide bridges formation more than abnormal protaminization, we used E_2_ treatment as an experimental approach useful for recovery sperm protamination efficiency in *CB1*-*null* mice and verify the impact of CB1 deletion on chromatin maturation of efficiently-protamined *CB1*-*null* SPZs during the epididymal transit.

The experimental control confirmed E_2_-treatment efficiency. The abnormal histone retention observed in CB1^-/-^ testis was responsive to treatment as E_2_ restored testicular content of histone H3 to WT values. This rescue activity on histone displacement was sufficient to predispose release in *caput* epididymis of efficiently protamined and condensed SPZs, which includes HDS and TDS values comparable in *caput* SPZs from WT mice and E_2_-treated mice. This also confirms previous results [38]. However, in all the experimental groups, SPZs showed high HDS and TDS values, in both *caput* and *cauda* epididymis. While HDS values significantly increased from *caput* to *cauda*, TDS values remained stably elevated. This was observed independently of treatment and in agreement with the genotype of treated animals revealing that failure of protamine thiol oxidation occurred, which was in agreement with *CB1*-gene deletion more than sperm protamine content. Likely, the observed recovery of histone displacement and sperm histone/protamine content was not sufficient to limit chromatin swelling during the epididymal transit in the absence of CB1. We concluded that inter/intra-protamine disulphide bonds formation is responsive to CB1. We are not able to assert that CB1 locally modulates epididymal sperm maturation. However, our data reveal CB1 involvement in packaging of sperm chromatin during the epididymal transit. Mouse and human SPZs express CB1 and produce endocannabinoids [24] and, more remarkably, a recent study carried out in humans describes CB1 in epididymal cells [53]. These are all findings suggesting that CB1 may potentially affect chromatin maturation of SPZs by local action. Further analyses need to verify this aspect and assess the endocrine and/or local control exerted by CB1. The data reported above is in agreement with the previously mentioned references showing that histone displacement and inter/intra-protamine disulphide bonds formation are both responsive to CB1.

To fulfill some gaps of knowledge about the neuroendocrine role of CB1 in the histone displacement event, using WT and CB1^-/-^ mice, we characterized the intra-testicular deficit of E_2_ levels [33] and molecular mechanism associated to abnormal histone retention. In particular, we analyzed gene expression of key modulators such as *Brdt* and *Cdyl* [10,11], and levels of histone H4tetraAc levels as the main signal of displacement.

In agreement with aromatase expression and physiological levels of testicular E_2_ [33,54], results showed that CB1 gene deletion reduced intra-testicular estrogens levels. E_2_ levels decreased by ~40% in CB1^-/-^ vs. WT testis. Such a decrease was sufficient to affect histone removal since histone H3 content was higher in CB1^-/-^ mice compared to WT both in testis and SPZs. This abnormal histone H3 retention was associated with no change of gene expression of *Brdt* or *Cdyl* while H4tetraAc decreased significantly in CB1^-/-^ compared to WT demonstrating that *CB1* deletion affected histone displacement by interfering with hyperacetylation of histone H4. It is easy to speculate that 40% of E_2_ reduction was sufficient to interfere with the histone mark of displacement, predisposing the abnormal histones retention in both testis and SPZs. Although several studies report findings about the role of E_2_ in SPT differentiation [2], no information is available about the intra-testicular E_2_ levels accountable for inefficiency of histone displacement.

In conclusion, our results show that CB1, via paracrine and/or endocrine cell-to-cell signaling, modulates chromatin remodeling mechanisms that orchestrate nuclear condensation extent of mature SPZs. We show that *CB1*-gene deletion affects the epididymal phase of chromatin condensation by interfering with inter-/intra-protamine disulphide bridges formation. In addition, it deranges efficiency of histone removal by reducing the hyperacetylation of histone H4.

Our results reveal a novel and important role for CB1 in sperm chromatin condensation mechanisms. Any interference with CB1 activity, including the marijuana consumption, might interfere with a sperm chromatin maturation status with potentially damaging effects on sperm quality.

## Figures and Tables

**Figure 1 ijms-21-03117-f001:**
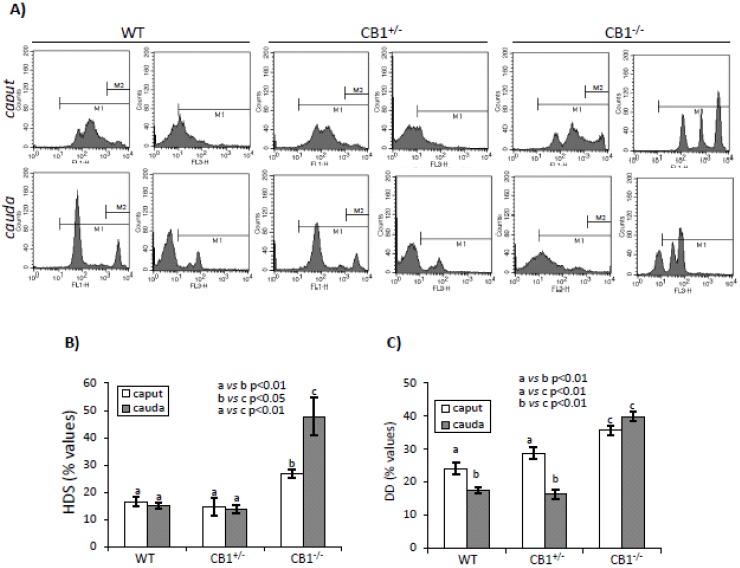
Flow cytometry analysis of sperm from *caput* and *cauda* epididymis of WT, CB1^+/-^, and CB1^-/-^ stained with AO. (**A**) Representative histograms of AO stained sperm in M1and M2 gate. Intensely green (FL1-H >10^3^), green (FL1-H >10^1^), red (FL3-H >10^1^) and total (green + red) fluorescencing DNA were used to analyze: (**B**) HDS (used as index of uncondensed chromatin) and (**C**) DD (used as index of DNA damage) values. Graphs were representative of four sperm samples/animal/genotype (*n* = 4 animals for each genotype in triplicates). Data were expressed as the mean values ± S.E.M. Different letters indicated statistical significance (*p* < 0.05 or *p* < 0.01).

**Figure 2 ijms-21-03117-f002:**
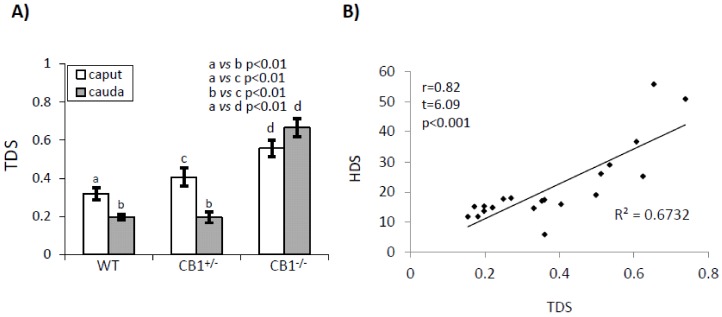
Flow cytometry analysis of sperm from *caput* and *cauda* epididymis of WT, CB1^+/-^, and CB1^-/-^stained with AO. Green (FL1-H > 10^1^) and red (FL3-H >10^1^) fluorescent DNA were used to analyze: (**A**) TDS (used as index of thiol groups oxidation) values. Graphs were representative of four sperm samples/animal/genotype (*n* = 4 animals for each genotype in triplicates). Data were expressed as the mean ± S.E.M. Different letters indicated statistical significance (*p* < 0.05 or *p* < 0.01). (**B**) Correlation analysis between HDS and TDS values relative to *caput* and *cauda* SPZ of WT, CB1^+/-^ and CB1^-/-^, independently by genotype and epididymal region (*r* = 0.82, *p* < 0.001).

**Figure 3 ijms-21-03117-f003:**
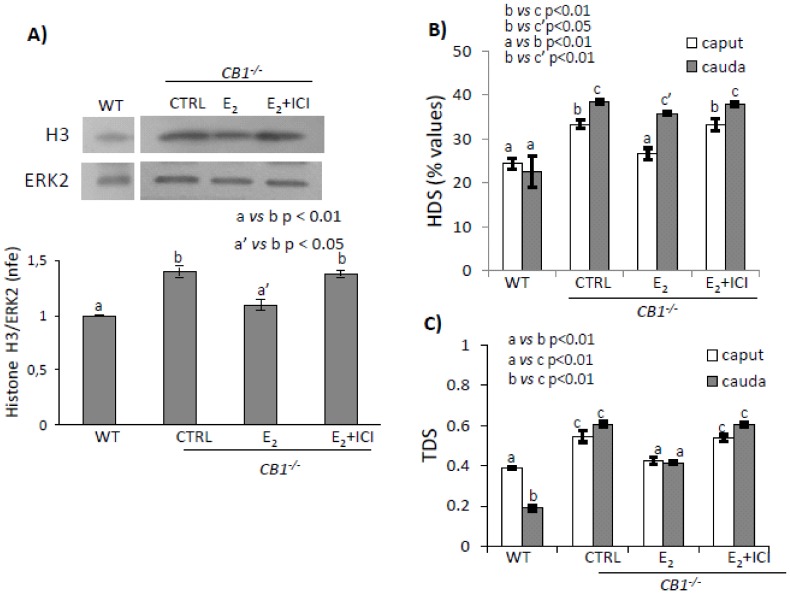
(**A**) Western blot analysis of Histone H3 in testis from WT and CB1^-/-^ mice in vivo treated with vehicle (CTRL) or E_2_ ± ICI. H3 amount was quantified by densitometry analysis, normalized against ERK2 signals, and expressed as a fold change of OD values. Graphs were representative of or four testis/experimental group (*n* = 3 or 4 testes from different animals for each experimental group in triplicates). Flow cytometry analysis of *caput* and *cauda* SPZ of CB1^-/-^ mice in vivo treated with vehicle (CTRL) or E_2_ ± ICI. (**B**) HDS (used as index of uncondensed chromatin) and (**C**) TDS (used as index of thiol groups oxidation) values. Graphs were representative of three or four sperm samples/animals/experimental groups (*n* = 3 or 4 animals for each experimental group in triplicates). All data are reported as mean values ± S.E.M. Different letters indicated statistical significance (*p* < 0.05 or *p* < 0.01, a’ vs. a: not significant).

**Figure 4 ijms-21-03117-f004:**
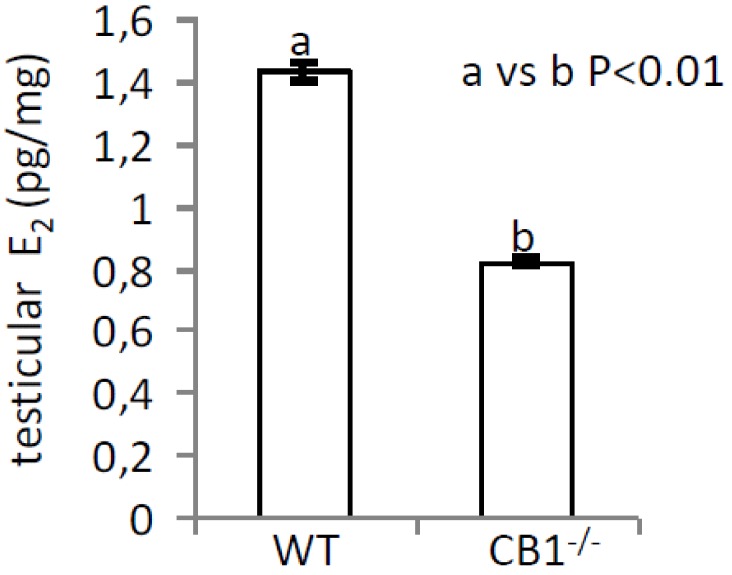
E_2_ levels in testis from WT and CB1^-/-^ mice by EIA assay. Values were expressed in pg for mg of tissue (pg/mg) and reported as mean values ± S.E.M. Graph was representative of three testis from six different animals/genotype (*n* = 6 animals for each genotype in triplicates) (a vs. b *p* < 0.01).

**Figure 5 ijms-21-03117-f005:**
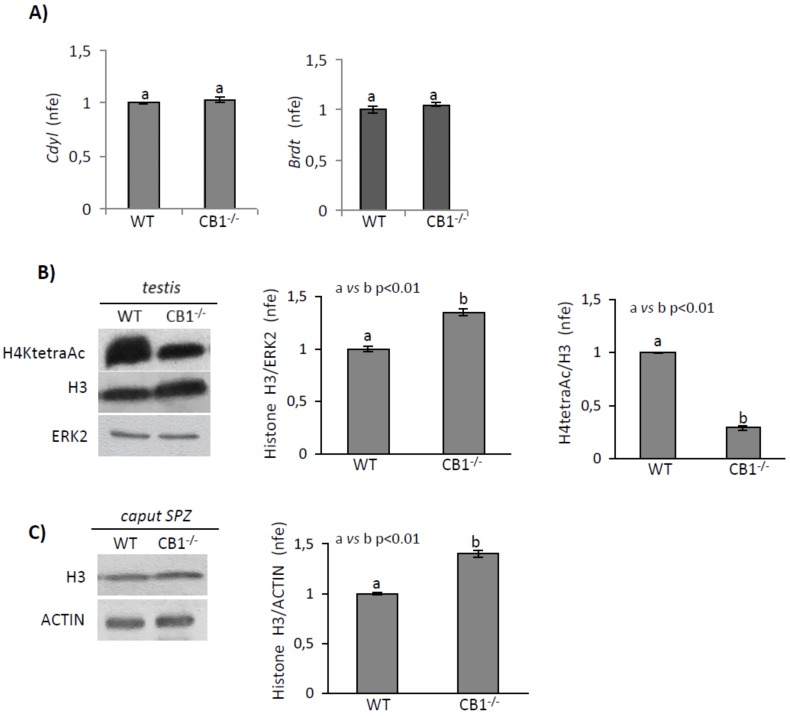
RTqPCR analysis of *Cdyl* and *Brdt i*n testis of WT and CB1^-/-^ mice (**A**). Transcript amounts was reported as normalized fold expression (nfe) relatively to *Rps18* gene. Western blot analysis of (**B)** Histone H4tetraAc and Histone H3 in testis and (**C**) Histone H3 in *caput* SPZ of WT and CB1^-/-^ mice. H4tetraAc and H3 amount was quantified by densitometry analysis, normalized against ERK2 (for testis) or ACTIN (for *caput* SPZ) signals and expressed as a fold change of OD values. All data are reported as mean values ± S.E.M. Graphs were representative of six or seven testis from six/seven different animals/genotype (*n* = 6 or 7 animals for each genotype in triplicates). Different letters indicated statistical significance (*p* < 0.05 or *p* < 0.01).

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
