# Peer review of "The Cannabinoid Receptor CB1 Stabilizes Sperm Chromatin Condensation Status During Epididymal Transit by Promoting Disulphide Bond Formation"

_ijms, 2020, doi:10.3390/ijms21093117_

Round 1

Reviewer 1 Report

figure 2B still does not state from which region of the epididymis the data shown is generated.  is the HDS/TDS ratio shown from the caput or cauda?

Author Response

REVIEWER 1

Comments and Suggestions for Authors

figure 2B still does not state from which region of the epididymis the data shown is generated.  is the HDS/TDS ratio shown from the caput or cauda?

We thanks reviewer 1 for his consideration. We modified manuscript accordingly. In the 2.8. section of Material and Methods, we specified that for Correlation Analysis in the figure 2B, all the HDS and TDS values relative to caput and cauda SPZ of WT, CB1+/- and CB1-/- mice have been correlated each other. Therefore, we included in our analysis both caput and cauda SPZ without consider the genotype and epididymal region (i.e. independently by genotype and epididymal region). We added the same informations also in the figure 2B legend.

Reviewer 2 Report

The authors investigate the effect of CB1 on sperm 3 chromatin condensation.

A significant concern is the low n value. n =3 seems to not be sufficiently powered, and the data would not be normally distributed to justify the statistics

Minor Comments:

Abstract:

Define CDYL and  BRDT

Introduction

Define CDYL and BRDT

PAGE 2, line 60-should remove the word anyway

Reference should be cited: All the events above described are finely regulated by neuroendocrine axis and local factors that 80 act through endocrine, autocrine, and paracrine pathways

Methods:

Are CB1 KO on CD1 background?

Why is the age range 4-8 months? Could age have affected the outcomes from this study?

n=3-4-is this sufficiently powered?

Why this dose of E2?

remove properly line 151

define AO

Were the data normally distributed to justify the statistics used?

Results:

n value required in figure legends

Discussion

remove this: (see WT vs CB1-/- 424 treated mice)

remove For what we know,

Author Response

Comments and Suggestions for Authors

The authors investigate the effect of CB1 on sperm  chromatin condensation.

A significant concern is the low n value. n =3 seems to not be sufficiently powered, and the data would not be normally distributed to justify the statistics

We thank reviewer 2 for his considerations. To answer about the number of animals used in this study, it must be said that in Italy all research projects involving the use of animals must be authorized by the Ministry of Health,  the competent  authority, that issues a specific authorization after an evaluation and acquired the opinion of other technical-scientific bodies.

In this regard, there is a very stringent legislation on the number of animals to be used in a research project, which takes into account, among the various principles, also that of the maximum reduction in the number of animals used, compatible with the objectives of the research project.

The approving body dictates statistical parameters (statistical power) to describe how the number of animals necessary for the study was determined.

The animal experiments described in this work is part of a much larger ministerial project (authorization n ° 941/2016-PR issued on 10.10.2016) and at the time of the request for approval, to calculate and justify the number of animals enrolled, a G-Power analysis was used, based on preliminary data available at the time.

The number of animals used for each genotype or experimental group has been fixed by G*Power analysis using G*Power 3.1.9.2 software which  suggested at least 3 animals/group or genotype to have 0.96 as power calculation, being P value  fixed to 0.05.

The number of the animals enrolled in the study in some experimental procedures was slightly exceeding the one fixed by the statistical analysis by means G*Power software. To have an useful power calculation (i.e. actual power 0.96) we used the following parameters:

3.48 effect size 

0.05 (a error probability)

0.8 power (1- b err prob)

The use of these parameters returned that group should be formed by at least 3 separate animals to have 0.96 as power calculation. Therefore, 3 or 4 animals/genotype or experimental group were analyzed. In the new version of the Manuscript, detailed information about the number of animals enrolled has been added including power calculation.

Anyway, since we are continuing to study on the impact of CB1-gene deletion on histone displacement mechanism we have continued to collect data. Therefore, in the new version of the Manuscript, where possible, we have implemented the number of animals and observations considered. The text, figure legends and in particular graphs of  Figures 4 and 5 have been changed, accordingly.

Minor Comments:

1) Abstract:  Define CDYL and  BRDT

We changed the Abstract, accordingly.

2) Introduction

- Define CDYL and BRDT

- PAGE 2, line 60-should remove the word anyway

- Reference should be cited: All the events above described are finely regulated by neuroendocrine axis and local factors that act through endocrine, autocrine, and paracrine pathways

We changed the Introduction section, accordingly.

3) Methods:

-Are CB1 KO on CD1 background?

Yes, CB1 KO male mice are on CD1 background. In fact, in the 2.1. section of Material and Methods, we described how CB1 KO mice are generated; in particular, we specified that heterozygous mice (CB1+/-) were bred on a CD1 background and after crossed (keeping two lines of separate progenitors) to generate male mice (WT, CB1+/- and CB1-/-) used in this study.

-Why is the age range 4-8 months? Could age have affected the outcomes from this study?

We decided on the age of animals based on our previous study concerning the possible role of CB1 receptor in chromatin remodelling. We have changed the text inserting references, accordingly.

-n=3-4-is this sufficiently powered?

Yes. The number of animals used for each genotype or experimental group has been fixed by G*Power analysis using G*Power 3.1.9.2 software.

-Why this dose of E2?

We decided on doses and time of treatment based on our previous studies demonstrating the recovery of sperm protamination efficiency in CB1-null mice by E2 treatment.

Here, using this experimental approach (type of administration, doses and time of treatment) we studied the impact of CB1-gene deletion on sperm protamine thiol group oxidation during the epididymal transit, by avoiding interference by poor protamination.

We have changed the text inserting references, accordingly.

-remove properly line 151

We have changed the text, accordingly.

-define AO

We have changed the text, accordingly.

-Were the data normally distributed to justify the statistics used?

Each variable was tested for normality and  homogeneity of variance was assessed. Since the distribution of the variables was normal and the values were homogeneous in variance, all statistical analyses were performed using one-way analysis of variance (ANOVA) followed by Duncan’s test (for multigroup comparison) or Student’s t-test (for two independent group comparison).

4) Results:

  • n value required in figure legends

We have changed the figure legends inserting n value, accordingly.

In addition, in accord to the recommendations of the same reviewer, we have implemented the number of observations and animals considered, where possible and consequently we have modified the graphs of figures 4 and 5.

5) Discussion

-remove this: (see WT vs CB1-/- 424 treated mice)

-remove For what we know

We changed the Discussion section, accordingly.

Round 2

Reviewer 2 Report

No additional comments

This manuscript is a resubmission of an earlier submission. The following is a list of the peer review reports and author responses from that submission.

Round 1

Reviewer 1 Report

The manuscript entitled « The cannabinoid receptor CB1 stabilizes sperm chromatin condensation status during epididymal transit by promoting disulphide bond formation » by Chioccarelli and colleagues presents a series of experiments performed on a global CB1 knock-out mouse model that sustain eventual new findings regarding the role of CB1 and on sperm chromatin condensation. While these experiments seem to be overall well conducted, the insights gathered from these do not appear to significantly improve our knowledge regarding the direct implication of the endocannabinoid pathway in male germ cell physiology as compared to previous studies, notably from the same research group. Actually the data presented in this manuscript more or less recapitulate previous results but using slightly different readouts or different technics. Eventually, the only new information from this study is that the altered chromatin condensation of sperm cells persists after they transited through the epididymis. There is however no functional data to identify whether this results from the altered function of the epididymis ducts, or whether this is just the logical consequence of the already altered chromatin condensation in these cells.

Major and general comments:

1) The alteration of chromatin condensation in sperm cells from the caput epididymis was already reported. Most, if not all conclusions from this study are in agreement with previous ones from the same group (e.g. refs 27, 28 and 34), which makes it very redundant. The main reason for this that the endpoints that were analysed and the means to do so are very similar to what was previously done. For instance:

- Only AO staining was used in this study, and this had been performed previously in combination with aniline blue or PI staining, to measure the exact same parameters, i.e. HDS and DD values; Only the TDS parameter appears to be new, but it is again indirectly measured by AO staining. This should/could have been included in previous studies.

- Here, the intra-testicular levels of 17 b-estradiol were found to be decreased but this is completely in line with the decreased plasma levels and the down-regulation of intra-testicular Cyp19a1 mRNA levels that were already reported in this model.

- In this study the levels of histone H3 protein levels were monitored and were found to be increased, while previously the levels (mRNA and/or protein) of TNP1/2 and PRM1/2 were found to be decreased, which does not add much to (or eventually should have been done in previous) studies, where it was found that CB1 affects Histone displacement” and its KO interfers with “chromatin organization”;

- The effects of E2 and E2 + ICI treatments related treatments are essentially the same as what was previously reported, bur on slightly different endpoints, measured by relatively similar means. Overall it therefore shows what was previously showed, i.e. E2 restores or partially restores the affected parameters.

2) Eventually, the new phenotypic characterisation of this model consists in the sperm cell status regarding disulphide bonds, for which evaluation before and after epididymis transit, with or without E2 or and related treatments seem to indicate - according to the authors - a direct involvement of CB1 (Figure 6). The demonstration for this is however quite indirect as already mentioned (Red fluorescing versus Green fluorescing) but more importantly the corresponding results are not convincing. First, the absence of appropriate controls (WT samples) in Figures 6A and 6B makes interpretation difficult. Second, the absence of any differences at all between caput and cauda data in figure 6B, especially for “CRTLS” (untreated CB1-/- samples) whereas TDS values were different for these same samples in Figure 2A (although the differences were not statistically significant) is intriguing and raises serious concerns about these results. Therefore conclusions regarding the fact “the failure of disulphide bridges formation in SPZ transiting from caput-to-cauda was related to deficit in CB1 activity (i.e. deletion) during the epididymal transit” are questionable. In the end it cannot be concluded about the role of the epididymis in this regards, and it remains very likely that the altered chromatin condensation of SPZ in the cauda is the simple consequence that this was already the case before the transit. And once again this “caput” phenotype was already described.

3) The global CB1 knock-out model is rather complex, especially when it comes to fertility. Indeed in this model the HPG axis appears significantly altered which results (unless this is the other way around) in alterations of both gonadotropin (Hypothalamic Gnrh, pituitary Gnrhr and Fshb) and intragonadal hormone levels. The functional explanations regarding the way CB1 would act on sperm cell chromatin solely resides on the involvement of E2. First, this does not add much to previous knowledge (see comment #1). Second, is there a specific alteration that would not be explained by the dysregulation of E2 levels? The fact that CB1 regulates (directly or indirectly) E2 levels and that this alters sperm chromatin is very clear (and was already demonstrated by the same group) but what is the point here: To demonstrate regulatory role of E2 on spermatogenesis (even though this is still interesting) or that of endocanabinoids? On top of that, as indicated by the authors in the Material and Methods section, treatments with ICI have deleterious effects and testis and associated ducts (which justifies the absence of ICI treatments alone as a control), its use is eventually questionable. Furthermore, while E2 treatments on CB1-/- most of time restore the analysed parameters to apparently normal values, it would be interesting to evaluate the effects of this treatment in the absence of CB1 deletion, i.e. on WT mice. Altogether, this indicates that a cleaner model (e.g. germ cell specific or gonadal specific knock out) is now needed to solve these issues and get more insights into the regulatory role of CB1 and endocannabinoids in male fertility.

Other minor and more specific comments:

1) Since HDS and TDS values somehow rely on related measurements (i.e. red and / or green fluorescence) is it not normal to find a correlation between values? What is the biological interpretation of this correlation? Is it the case also for DD? The DD profiles being highly similar those for TDS, it is tempting to speculate that these would be correlated too.

2) Cyp19a1 mRNA levels are decreased in this model, which likely explains the decrease of intra-testicular levels of E2. Since i) the HPG axis and testicular hormones are affected in this model, and ii) aromatisation of T by aromatase (encoded by Cyp19a1) is the source of intra-testicular E2, it is tempting to ask about the intra-testicular levels of T in this model?

3) What is the rationale for using different loading controls for testis (ERK2) and caput SPZ (ACTIN) in figure 4?

4) Figure 6: Data for WT samples should be included as a control in order to properly evaluate the effect(s) of treatments. This is relatively easy since data already exist (Figure 1B and Figure 2A)

5) All figures: The number of individuals should be included in each legend. It would prevent the reader from going back to the material and method section each time.

Reviewer 2 Report

The work presented is an extension of previous studies focusing on the potential role(s) of CB1 in chromatin condensation in epididymal sperm.  However, there are a few modifications and clarifications that should be addressed as indicated below:

1) in the stats section is states the mean is from 3 independent samples.  For sperm samples is this 3 samples from 1 animal, 1 sample from each of 3 animals, or 3 samples from 3 animals?  

2) In figure 2A, the significance of a vs d should also be included.  

3) What region of the epididymis was used to generate data in figure 2B?

4) Figure 3 should be a bar graph since the x-axis data in not contiguous.

5) in figures 3 and 4, why was data from the CB+/- group not included as is was in figures 1 and 2?  Please include this data or give a rationale for its exclusion.

6) In figure 4, why is actin used as the western blot loading control for sperm and ERK2 used for the tests?  Please explain this choice in the methods or results section.  

7) In figure 5, the p value of a vs a' should also be included.

8) In the results section (e.g. lines 255-256, 275-276) in states animals were injected with vehicle or E2+ICI.  animals were also injected with E2 and should be included in these sentences.  

9) Injections were started at pnd 24 and covered the initial wave of spermatogenesis.  Why was the first wave chosen?  given the potential uniqueness of the first wave of spermatogenesis, why not perform in older animals?